# The impacts of recent drought in lowland Bolivia on fire, forest loss and regional smoke emissions

Joshua P. Heyer[1], Mitchell J. Power[1,2], Robert D. Field[3,4], and Margreet J.E. van Marle[5,6]

[1]Geography Department, University of Utah, Salt Lake City, UT 84112-9155, USA
[2]Natural History Museum of Utah, University of Utah, Salt Lake City, UT 84112, USA
[3]NASA Goddard Institute for Space Studies, New York, NY 10025, USA
[4]Department of Applied Physics and Applied Mathematics, Columbia University, New York, NY 10025, USA
[5]Faculty of Earth and Life Sciences, Vrije Universiteit Amsterdam, Amsterdam, Netherlands,
[6]Deltares, Delft, the Netherlands

*Correspondence to*: Joshua P. Heyer (josh.heyer@geog.utah.edu)

**Abstract**. In the southern Amazon relationships have been established among drought, human activities that cause forest loss, fire, and smoke emissions. We explore the impacts of recent drought on fire, forest loss, and atmospheric visibility in lowland Bolivia. To assess human influence on fire, we consider climate, fire and vegetation dynamics in an area largely excluded from human activities since 1979, Noel Kempff Mercado National Park (NK) in northeastern Bolivia. We use data from five sources: the Moderate Resolution Imaging Spectroradiometer Collection 6 active fire product (2001–2015) (MODIS C6), Global Fire Weather Database data (1982–2015) (GFWED), MODIS land-cover data (2001–2010), MODIS forest loss data (2000–2012), and the regional extinction coefficient for the southwestern Amazon (i.e., $B_{ext}$), which is derived from horizontal visibility data from the World Meteorological Organization (WMO)-level surface stations (1973–2015). The $B_{ext}$ is affected by smoke and acts as a proxy for visibility and regional fire emissions. In lowland Bolivia from 2001–2015, interannual Drought Code (DC) variability was linked to fire activity, while from 1982–2015, interannual DC variability was linked to $B_{ext}$. From 2001–2015, the $B_{ext}$ and MODIS C6 active fire data for lowland Bolivia captured fire seasonality, and covaried between low and high fire years. Consistent with previous studies, our results suggest $B_{ext}$ can be used as a longer-term proxy of regional fire emissions in southwestern Amazonia. Overall, our study found drought conditions were the dominant control on interannual fire variability in lowland Bolivia, and fires within NK were limited to the cerrado and seasonally-inundated wetland biomes. Our results suggest lowland Bolivia tropical forests were susceptible to human activities that may have amplified fire during drought. Human activities and drought need to be considered in future projections of southern Amazonia fire, in regard to carbon emissions and global climate.

## 1 Introduction

Observations from the southern Amazon reveal fire emissions increased from 1987 through the early 2000s (van Marle et al., 2017). During this time, humans used fire in the southern Amazon while logging timber, and to clear land for building infrastructure and agriculture (Moran, 1993; Nepstad et al., 1999; Fearnside, 2005; Morton et al., 2008; Cochrane, 2009; Nepstad et al., 2009; van der Werf et al., 2010), suggesting human activities had a significant impact on increased smoke emissions from fire (van Marle et al., 2017). To minimize the impacts of deforestation and fire on deforestation in the

Amazon, and in turn on carbon emissions and global climate, restrictions on land expansion and policies regulating beef and soy production in the Brazilian Amazon were implemented during the early 2000's (Nepstad et al., 2014). While these restrictions and policy changes helped reduce deforestation from 2004 – 2013 (Nepstad et al., 2014), others suggest a decrease in demand for Amazon resources was the primary driver of reduced deforestation and fire from 2004 - 2012 (Fearnside, 2017). Outside of the Brazilian Amazon, forest loss from deforestation and fire has continued in parts of the

southern Amazon, particularly in lowland Bolivia (Chen et al., 2013b; van Marle et al., 2016; van Marle et al., 2017). The cerrado biome in particular has experienced increased deforestation since 2010, which could be due to a shift in agricultural to the southern Amazon and cerrado (Soares-Filho et al., 2014). Likely amplifying the effects of deforestation and fire on forest loss in the southern Amazon were drought conditions during the early 2000s (Brown et al., 2006; Aragão et al., 2007; Marengo et al., 2008), raising the question, what are the relationships among recent drought, fire and forest loss in lowland

Bolivia?

      Both paleofire investigations (Bush et al., 2008; Marlon et al., 2008; Power et al., 2013) and modern fire records (Brown et al., 2006; Aragão et al., 2007; Marengo et al., 2008) link drought to fire in the southern Amazon. Our study considers relationships between recent drought and fire in lowland Bolivia using the Global Fire Weather Database (GFWED) (Field et al., 2015) and the Moderate Resolution Imaging Spectroradiometer collection 6 (MODIS C6) active fire product (Giglio et

al., 2016). The GFWED Drought Code (DC) in particular captures net drying of deep fuels, with lower DC values observed during the wet season and higher DC values during the dry season (Field et al., 2015). We interpret high (low) DC values during the fire season from August–October in lowland Bolivia as an indicator of antecedent dry (wet) conditions during the preceding wet and dry seasons.

In addition to understanding links between drought and fire in lowland Bolivia, we also consider where past fires occurred spatially in relation to land use and biome type using data from the MODIS-based global land cover product (Broxton et al., 2014), and the Landsat-based forest loss product (Hansen et al., 2013). Considering humans have had a significant impact on forest loss and fire activity in unprotected biomes, we compare fire distribution in unprotected biomes in lowland Bolivia to biomes in Noel Kempff Mercado National Park (NK), an area in lowland Bolivia protected from deforestation since 1979. Specifically, we explore climate and fire relationships in NK to determine when fire activity is high in relation to interannual climate variability, and where fire occurs spatially in relation to different biomes and land uses.

Finally, to extend our fire record for lowland Bolivia prior to the onset of MODIS C6 in 2001, we analyze horizontal visibility data from World Meteorological Organization (WMO) level surface weather stations. Visibility data has been used as a fire emissions proxy to understand fire activity over the southern Amazon from 1973–2015 (van Marle et al., 2017). We test relationships between MODIS C6 active fire data for lowland Bolivia and regional WMO-visibility data (locations of weather stations in Fig. 1a), to determine how well visibility data corresponds to the MODIS C6 fire record from 2001–2015, and to extend the fire record for lowland Bolivia prior to 2001.

Our results are useful and relevant when considering uncertainties regarding the fate of the southern Amazon in response to climate change (Zhang et al., 2015). Here, we show how recent interannual climate variability has impacted fire activity across different biomes in lowland Bolivia. A further understanding of relationships between interannual climate variability, biome type and fire in lowland Bolivia is valuable when considering fire weather (Bedia et al., 2015) and fire-season severity are expected to increase in the southern Amazon during the 21$^{st}$ century (Flannigan et al., 2013). Further, fire in the southern Amazon can cause increased smoke emissions that negatively impact human health (e.g., Brown et al., 2006), and can impact carbon emissions and global climate (Fearnside, 2005; Aragão et al., 2014). Seasonal covariation between fire and horizontal visibility data are explored here to provide further information on how fire is related to visibility and smoke emissions in the southern Amazon.

## 2 Methods and Data

### 2.1 MODIS C6 and Landsat Data

The Moderate Resolution Imaging Spectroradiometer Collection 6 active fire product (MODIS C6) offers a tool to

adequately answer questions related to recent fire activity from 11/2000–present (Morton et al., 2011). MODIS C6 has been

used globally to explore a variety of fire related questions ranging from biomass burning (Wooster et al., 2003) and fire

detection in the Amazon (Chen et al., 2013a). High-spatial resolution (1 kilometer) near real time MODIS C6 data used in

our analyses are provided by the Land, Atmosphere Near real-time Capability for Earth Observing System Fire Information

for Resource Management System, and operated by the NASA/Goddard Space Flight Center/Earth Science Data and

Information System (Giglio et al., 2016). Two key limitations of the previous MODIS C5 product were small forest clearings

causing false active fires, and thick smoke obscuring large fires (Giglio et al., 2016). For tropical ecosystems, these two key

limitations were addressed and errors were reduced from MODIS C5 to MODIS C6 (Giglio et al., 2016).

For our study, MODIS C6 data for Bolivia (Fig. 1b) was downloaded using the NASA Earth Observation Data archive

download tool. Data was subset by location (e.g., NK), year and by fire detection confidence ≥ 90%. Only active fire points

≥ 90% were included in our analyses to further reduce the potential for false active fires. MODIS C6 data seen in Brazil are

shown in spatial analyses (i.e., Fig. 4 and 5), but were not used in statistical analyses (Tables 1, 2, 3).

Landsat forest loss data 2000–2012 (Hansen et al., 2013) and MODIS based Collection 5.1 MCD12Q global land cover

data (Broxton et al., 2014) were obtained to determine the spatial coherency of fire, land use, and forest loss. High-spatial

resolution figures (i.e., Fig. 1, 4, 5) were created for detailed spatial analyses. Considering the high-spatial resolution of

certain figures, to view forest loss displayed as white pixels (i.e., Fig. 5), or detailed biome and fire spatial variability (e.g.,

Fig. 1), readers will need to increase the zoom. To simplify in-text discussions on the spatial distribution of fire in lowland

Bolivia in relation to various biomes, MODIS land-cover types seen in figure legends (i.e., Fig. 1b, 4, 5) will be referred to

in the paper hereafter as the cerrado, METF, SDTF, and seasonally-inundated wetlands. The cerrado biome includes open

shrublands, woody savannas, savanna, and grasslands MODIS land-cover types. The SDTF biome includes deciduous

broadleaf forest, mixed forest, and closed shrublands MODIS land-cover types. The METF biome includes the evergreen

broadleaf forest MODIS land-cover type. Seasonally-flooded wetlands in lowland Bolivia are hydromorphic climatic

savannas that are periodically flooded during the wet season and desiccate during the dry season (Junk et al., 2011). For

readers interested in the more detailed land-cover classification (Broxton et al., 2014), the original land classification was maintained in figures (i.e., Fig. 1b, 4, 5) for detailed spatial analyses of fires and biomes over lowland Bolivia.

## 2.2 Horizontal Extinction Coefficient calculated from Visibility Observations

In the absence of long-term fire data, horizontal visibility has been used as a proxy for fire emissions in Indonesia (Field et al., 2009; Field et al., 2016) and Amazonia (van Marle et al., 2017). Here, horizontal visibility observations (1973–2015) were taken from the NOAA National Climatic Data Center Integrated Surface Database (https://catalog.data.gov/dataset/integrated-surface-global-hourly-data), which is comprised of data from WMO-level surface stations provided by national meteorological agencies. Horizontal visibility observations are human-made using landmarks with known distances during the day and using point light sources at night (World Meteorological Organization, 1996). Horizontal visibility is influenced by several sources including dust, air pollution, haze, fog, and precipitation. Fires also have a strong impact on visibility, and therefore is used here as a proxy of regional fire emissions in lowland Bolivia.

To correct for limitations of the human eye and imperfections of the landmarks used to estimate the maximum distance seen, the observations in meters are usually expressed as the extinction coefficient ($B_{ext}$, $km^{-1}$). In our case, $B_{ext}$ corresponds to the degree to which light is attenuated by scattering and extinction over a horizontal path. Eleven stations were selected and the monthly median $B_{ext}$ over these selected stations was found representative for fire emissions over the region (7°S–17°S, 58°W–68°W) (Fig. 1a). We only used individual observations taken at 12:00 UTC (corresponding to 08:00 local time), as the frequency of reports at other times varied considerably during the length of record. The data were subsequently filtered following Husar et al. (2000) and van Marle et al. (2017) to omit observations influence by fog or precipitation, and aggregated to monthly values.

## 2.3 Global Fire Weather Database

To examine climatic controls on fires in NK, we used weather parameters and components of the Fire Weather Index (FWI) System, which integrates different surface weather parameters influencing the likelihood of fires starting and spreading. The FWI System consists of moisture codes for three generalized fuel classes and three fire behavior components, computed each day from surface temperature, relative humidity (RH), wind speed and precipitation. Because of its flexibility, it is the most

widely used such system in the world, and has been adapted for use in different fire environments operationally and for research purposes (de Groot and Flannigan, 2014).

To explore interannual climate variability related to fire activity in lowland Bolivia, FWI data for period 1982–2015, were obtained from the Global Fire Weather Database (GFWED) (Field et al., 2015), and processed for a 50 km x 50 km bounding box over NK (13°S - 15.3°S, 62.2°W - 59.5°W ), and for a bounding box over lowland Bolivia. The GFWED,

gridded to a 0.5° latitude x 2/3° longitude resolution, includes different versions that are all computed using temperature, RH and wind speed from the NASA MERRA2-reanalysis (Gelaro et al., 2017), but using different estimates of daily precipitation, which is the most uncertain input to the FWI system, particularly in the tropics (Field et al., 2015). From these, we use in our analyses the Drought Code (DC), along with precipitation (mm day$^{-1}$), temperature (C°), and RH (%). The DC is an indicator of seasonal drying (Field et al., 2015), and is the simplest of the six FWI components. DC values that exceed

425 are considered extreme (Field et al., 2015). Precipitation, temperature and RH were included to compare the explanatory power of basic surface weather variables compared to the DC. For the DC and precipitation, we used versions computed from the 'raw' MERRA2 precipitation estimate and a MERRA2 rain gauge-corrected version used in the MERRA2 land surface scheme, to provide a measure of the dependence of our results on uncertainty in the precipitation estimates. Additional precipitation data were obtained from the Climate Prediction Center (CPC), the Global Precipitation Climatology

Project (GPCP), and the Tropical Rainfall Measuring Mission (TRMM). The CPC estimates precipitation from rain gauge data at 0.5° x 0.5° resolution (Chen et al., 2008), the GPCP estimated precipitation from satellites at 2.5° x 2.5° resolution (Huffman et al., 2009), and the TRMM estimated precipitation from satellites at 0.25° x 0.25° resolution (Huffman et al., 2007). For each GFWED variable, mean monthly time series were constructed for Bolivia and NK from 01/2001–12/2015, and from 01/1982–12/2015. Mean-fire season (August – October) time series were also created for selected GFWED

variables from 08/1982–10/2015.

### 2.5 Statistical Analyses

Several sets of linear correlations were performed in R to better understand seasonal and interannual relationships among MODIS C6, GFWED and WMO-visibility data. For each set of correlations, correlation coefficients were estimated using a

Pearson's coefficient, with a standard transformation to a t-statistic to assess significance (e.g., alpha level: 0.05). First, to

demonstrate seasonal covariation between fire and visibility, correlations were performed between monthly MODIS C6 data (i.e., total monthly fires) for lowland Bolivia and mean-monthly WMO-visibility (i.e., mean-monthly $B_{ext}$) data. Next, to provide information on seasonal relationships correlations were performed between monthly MODIS C6 data (i.e., total monthly fires) and mean-monthly GFWED data from 01/2001–12/2015, and between mean-monthly WMO-visibility data

and mean-monthly GFWED data from 01/2001–12/2015. Finally, to better understand interannual relationships, correlations for lowland Bolivia were performed between mean-fire season (i.e., August–October) GFWED data and mean-fire season WMO-visibility (i.e., mean-monthly $B_{ext}$) data from 1982–2015. In addition to Pearson's correlations, MODIS C6 fire climatologies were calculated for lowland Bolivia and NK for time period 2001–2015, to identify higher-than-normal fire years. WMO-visibility (i.e., mean-monthly $B_{ext}$) climatology was calculated for lowland Bolivia for time period 1983–2015,

to further demonstrate seasonal covariation between fire and $B_{ext}$ seasonality over lowland Bolivia.

## 3 Results

### 3.1 Fire spatial distribution and seasonality in lowland Bolivia and NK (2001–2015)

The spatial distribution of MODIS C6 active fires in lowland Bolivia and NK are seen from 2001–2015 (Fig. 1b). For

lowland Bolivia from 2001–2015, a significant relationship was found between mean-monthly MODIS C6 fire data and $B_{ext}$ data, with a 95% correlation confidence interval of 0.76–0.86. During this time 85% of mean-monthly MODIS C6 fires were from August–October, with an average of 10,574 fires/year (Fig. 2a). Both MODIS C6 fire seasonality and peak $B_{ext}$ occurred from August–October, demonstrating seasonal covariation. For NK from 01/2001–12/2015, 96% of mean-monthly MODIS C6 fires were from August–October, with an average of 65 fires/year (Fig. 2b).


### 3.2 Climate and fire relationships in lowland Bolivia

For lowland Bolivia from 2001–2015, higher-than-normal (i.e., > 10,574 fires/year) MODIS C6 fire years were identified in 2004, 2005, 2007, 2008, 2010, and 2011 (Fig. 3a). Relationships among Bolivia monthly MODIS C6 fire data and mean-monthly GFWED precipitation (Fig. 3b) and temperature (Fig. 3d) variables were weaker than relationships among Bolivia

monthly MODIS C6 fire data and mean-monthly GFWED relative humidity and DC variables (Table 1; Fig. 3c, e). However, statistical relationships among Bolivia monthly MODIS C6 fire data and mean-monthly GFWED temperature and

precipitation variables were still significant. Of all the GFWED variables, statistical relationships were strongest among mean-monthly MODIS C6 fire data and DC (Table 1), showing clear seasonal covariation (Fig. 3e).

Inverse relationships between MODIS C6 and precipitation across MERRA2, CPC, GPCP and TRMM were all comparably low (Table 1). Of the precipitation sources analyzed, the strongest observed relationship was between Bolivia MODIS C6 and MERRA 2 precipitation data, with a 95% correlation confidence interval of -0.27 – -0.51. A significant inverse relationship was observed between MODIS C6 active fire data and MERRA2 relative humidity, with a 95% correlation confidence interval of -0.56 – -0.73 for lowland Bolivia. Next, a positive relationship can be seen between MODIS C6 active fire and MERRA2 temperature data, with a 95% correlation confidence interval of 0.43 – 0.64. Finally, stronger relationships were observed among MODIS C6 active fire data and TRMM DC, GPCP DC, and CPC DC, when compared to correlations between MODIS C6 active fire data and MERRA2 DC. The strongest relationship for DC was between Bolivia MODIS C6 data and GPCP DC, with a 95% correlation confidence interval of 0.69 – 0.82.

### 3.3 Climate and fire relationships in NK

For NK from 2001–2015, higher-than-normal (i.e., > 65 fires/year) MODIS C6 active fire years were identified in 2003, 2004, 2005, 2007, 2010, and 2012 (Fig. 3f). Significant relationships were found among monthly NK MODIS C6 active fire data and mean-monthly GFWED variables for NK (Table 1). Relationships among monthly NK MODIS C6 active fire data and mean-monthly GFWED precipitation (Fig. 3g) and temperature (Fig. 3i) variables were weaker compared to relationships among monthly NK MODIS C6 active fire data and mean-monthly GFWED relative humidity and DC variables (Fig. 3h, j). The level of significance varied among monthly NK MODIS C6 active fire data and mean-monthly GFWED precipitation correlations.

Precipitation and fire inverse relationships were even lower over NK than over Bolivia across all different precipitation estimates (Table 1). The strongest observed relationship for precipitation was between NK MODIS C6 and MERRA 2 precipitation, with a 95% correlation confidence interval of -0.07 – -0.35. Next, a significant inverse relationship was observed between MODIS C6 active fire data and NK MERRA2 relative humidity, with a 95% correlation confidence interval of -0.21 – -0.47. A positive relationship was seen among NK MODIS C6 active fire and MERRA2 temperature data, with a 95% correlation confidence interval of 0.24 – 0.49. Finally, significant relationships were found among NK MODIS

C6 active fires and TRMM DC, GPCP DC, CPC DC, and MERRA 2 DC. Of the four DC, the MERRA2 DC had the weakest relationship to NK MODIS C6 active fires. The strongest relationship for DC was between MODIS C6 active fire data and

TRMM DC, with a 95% correlation confidence interval of 0.39 − 0.61.

### 3.4    Spatial distribution of fire and DC

During 2010, fire was widespread (Fig. 4a) and September GPCP DC values were higher and spatially coherent in northeast Bolivia (Fig. 4b). During 2014, fire was less active in lowland Bolivia (Fig. 4c) and GPCP DC during September was lower than 2010 (Fig. 4d). Outside of NK, fire occurred in the cerrado, SDTF, METF and seasonally-inundated wetland biomes

(Fig. 4a, c). Within NK, 223 MODIS C6 active fires were observed in 2010 (Fig. 4a), and 17 MODIS C6 active fires were observed in 2014 (Fig. 4c). During both years, fires in NK occurred primarily in the cerrado biome on the Huanchaca Plateau.

### 3.5    Fire and forest loss

Forest loss from 2000–2012, largely corresponded to areas where MODIS C6 fires occurred from 2001–2015 (Fig. 5). Within NK, the majority of forest loss can be found on the Huanchaca plateau where the cerrado biome is found, along the cerrado-METF boundary, and in seasonally-inundated wetlands. Overall, forest loss is minimal within NK compared to the unprotected areas adjacent to NK (Fig. 5). For unprotected areas outside of NK, forest loss occurred in the cerrado, SDTF,

METF, and seasonally-inundated wetland biomes, and in urban/agriculture land use zones (Fig. 5). Forest loss outside of NK largely corresponded to areas where fire also occurred (Fig. 5).

### 3.6 Seasonal and interannual relationships among $B_{ext}$ and GFWED data in lowland Bolivia

Overall, mean-monthly correlations among monthly MODIS C6 fire and mean-monthly GFWED variables (Table 1) were

stronger than mean-monthly correlations between $B_{ext}$ and GFWED variables from 2001–2015 (Table 2). In particular, correlations among $B_{ext}$ and GFWED precipitation and temperature variables were slightly weaker when compared to the same correlations using MODIS C6 fire data. Nevertheless, significant mean-monthly correlations were observed among $B_{ext}$ and GFWED DC and relative humidity from 1/2001–12/2015 (Table 2), suggesting seasonal covariation between $B_{ext}$ and DC.

235          Over the longer record from 1982–2015, significant correlations were observed among mean-fire season $B_{ext}$ and

GFWED data (Table 3). Strongest relationships were observed among mean-fire season $B_{ext}$ and GFWED DC and relative

humidity variables, with a tendency towards interannual covariation between $B_{ext}$ and DC (Fig. 6d). Relationships between

mean-fire season $B_{ext}$ and precipitation were significant, but overall weak. While interannual covariation is seen between $B_{ext}$

and temperature from 1982 – 1993 (Fig. 6c), the relationship between mean-fire season $B_{ext}$ and temperature was not

significant from 1982–2015 (Table 3).

## 4. Discussion

### 4.1 Drivers of fire in lowland Bolivia and NK

From 2001–2015, our analyses reveal strong fire seasonality in lowland Bolivia (Fig. 2a) and NK (Fig. 2b). Within NK,

drought conditions were the main driver of fire (e.g., Table 1; Fig. 3j), while fires in unprotected areas of lowland Bolivia

were controlled by a combination of drought (e.g., Figs. 3e, 4b, 6d), biome type (e.g., Fig. 5), and forest loss likely

influenced by human activities (Fig. 5). Driving drought conditions in lowland Bolivia are oceanic oscillations including El

Niño Southern Oscillation (Aragao et al., 2007; Bush et al., 2008; Asner & Alencar, 2010; Marengo et al., 2010; Grimm

2010), the Madden-Julian Oscillation (Marengo et al., 2010; Grimm, 2010), and Atlantic sea-surface temperature (SSTs)

variability (Vera et al., 2006; Aragao et al., 2007; Yoon & Zeng, 2010). Oceanic oscillations alter atmospheric circulation

associated with the South American monsoon (Vera & Vigliarolo, 2000; Vera et al., 2006b; Aragao *et al.,* 2007; Yoon &

Zeng, 2010; Marengo et al., 2011), causing precipitation deficits and region-wide drought in the southwestern Amazon

during the wet season (October/ November –April/May) (Aragao et al., 2007; Yoon & Zeng, 2010).

        High fire years in 2005, 2007 and 2010 (Figs. 3a, f), correspond to years of drought (Lewis et al., 2011; Chen et al.,

2013b) and high fire years in the southern Amazon identified by others (Chen et al., 2013b; van Marle et al., 2017).

Prolonged drought conditions in the southern Amazon are caused by reduced rainfall, higher-than-normal temperatures, and

reduced atmospheric moisture during the wet and dry seasons (Marengo et al., 2008). Our results suggest the CPC DC (Fig.

3e) and regional $B_{ext}$ (Table 2; Fig. 3a) captured interannual drought variability that impacted the southern Amazon during

the early 2000's. In addition to high DC values regionally, low relative humidity (Fig. 3c) and high temperature (Fig. 3d)

were linked to increased fire activity in lowland Bolivia and NK (Table 1). Further demonstrating the connection between

drought and fire in NK and lowland Bolivia were drought years and non-drought years, when fire was either significantly enhanced (Fig. 4a), or reduced (Fig. 4c). In particular, GPCP DC values were higher in lowland Bolivia during 2010, a high fire year (Fig. 4a, c), and GPCP DC values were lower during 2014, a low fire and DC year (Fig. 4b, d). Our results are consistent with others who found synchronous changes in fire activity between tropical forest and cerrado biomes in the

Amazon (Chen et al., 2013b). During 2014, fire in NK and the surrounding unprotected areas was significantly reduced across all biomes (Fig. 4b). The opposite occurred in 2010, when drought conditions and increased fire activity were observed across all biomes during the fire season (Fig. 4a, c).

While our results and paleosedimentary records (Burbridge et al., 2004; Maezumi et al., 2015) show fires are frequent on the cerrado landscape in NK (Killeen et al., 2002), determining if drought, human activities, or a combination of both

were the dominant drivers of recent fire activity in the SDTF and METF biomes in lowland Bolivia is less understood. Considering unfragmented tropical Amazon forests are more resilient to fire and drought conditions (Cochrane, 2003; Davidson et al., 2012) than fragmented tropical Amazon forests (Nepstad et al., 1999; Laurance and Williamson, 2001; Fearnside, 2005; Chen et al., 2013b), we suggest the spatial correspondence of forest loss and fire in METFs outside of NK (Fig. 5) was associated with forest clearing activities for economic practices common in the area (e.g., Killeen et al., 2008;

Fearnside, 2017).

Seen as forest loss (Fig. 5), the logging of Amazon forests has increased dry surface fuels, suppressed soil moisture, and created an environment susceptible to fire during drought (Nepstad et al., 1999; Fearnside, 2005; Chen et al., 2013a,b). Large geometric rectangular areas of forest loss observed in METFs and other biomes surrounding NK indicate forest loss from fire was not from lighting ignitions alone. While we did not directly monitor human activities, our results suggest human

activities further amplified forest loss and fire during high DC years. Spatial coherency between fire and forest loss seen in our results, and the results of others suggests fire in the SDTF and METF biomes is a function of human and lighting ignitions amplified by drought during the historical record (Killeen et al., 2002; Brown et al., 2006), and the paleorecord (Denevan, 1992; Bush et al., 2008). However, an area of fire and forest loss seen in the METF west of NK appears spatially random, and not necessarily because of human activities alone (Fig. 4a).

Spatial correspondence between fire, high DC and forest loss in the METF biome west of NK is observed in 2010 (Fig.

4a, c), indicating high DC (i.e., drought) was the dominant control on fire and forest loss. Drought was not limited to

lowland Bolivia in 2010. Rather, severe drought was observed across much of the Amazon basin (Lewis et al., 2011). The

drought in 2010, was linked to high Atlantic SSTs and intensified El Nino conditions (Lewis et al., 2011). While our

research focuses on small-spatial scale interactions between fire and local-to-regional climate variability, oceanic oscillations

impact drought in the Amazon (e.g., Aragao *et al.,* 2007), and likely influenced high fire years in lowland Bolivia during

2004, 2005, 2007, 2008, 2010, and 2011 (Fig. 3a).

## 4.2   Fire relationship to different biomes

Fire within NK is a function of biome type (Fig. 1b, 4, 5). In NK, fires ignited by lighting are frequent in the cerrado biome

from August-October (Killeen et al., 2002). Consistent with Cochrane (2003), a lack of fire activity was observed in the

METF biome in NK from 2001–2015. Fire during the MODIS C6 record in lowland Bolivia and in NK was largely restricted

to the cerrado and seasonally-inundated wetland biomes.

      Biome-boundary dynamics among the cerrado and other biomes influence fire in the southern Amazon (Power et al.,

2016). Extreme seasonal droughts can amplify the role of fire on biome-boundaries among the cerrado, SDTF, and METF.

The drying out of plant biomass and soil moisture increases the potential for cerrado grassland fire propagation into

neighboring biomes (Power et al., 2016). Amazon forest boundaries are vulnerable to positive feedbacks linked to forest loss

and climate induced drought that increases forest fragmentation and fire propagation (Laurance and Williamson, 2001).

Unprotected areas outside of NK show evidence of biome-boundary dynamics related to fire and forest loss (Fig. 5). The

majority of fires in the METF are observed at biome-boundary interfaces. Both boundary-biome dynamics and human

caused forest loss seem to have affected fire in the SDTF and METF biomes during out study.

## 4.3   Using visibility data as a proxy of interannual fire emissions in lowland Bolivia

We suggest $B_{ext}$ visibility data was a proxy of regional interannual fire emissions from 1982–2015. High mean-fire season

$B_{ext}$ in 1987, 1988, 1995, and 1999 could have been related to increased fire activity (Fig. 6). The five lowest visibility (i.e.,

high $B_{ext}$) years from 1982–2015, were observed towards the end of the visibility record in 1995, 1999, 2004, 2007, and

2010, corresponding to high MODIS C6 fire years in 2004, 2007, and 2010 (Fig. 3a). Our results are consistent with van Marle et al. (2017) who identified higher-than-normal particulate matter emitted into the atmosphere over the southwestern Amazon in 1988, 1995, 1999, 2004, 2007, and 2010. Consistency between our results and van Marle et al. (2017) suggests $B_{ext}$ visibility, with limitations in mind (van Marle et al., 2017), can be used as a proxy of regional fire emissions for the

southwestern Amazon and lowland Bolivia. Further supporting our findings are results showing fires in the Amazon cause the entrainment of smoke into the upper atmosphere, which enhance convective storms and lighting (Andreae et al., 2004). Correspondence between low visibility and high fire years in lowland Bolivia (Fig. 3a) suggests smoke emitted during high fire years could have enhanced convective storms and further amplified natural ignitions and fire.

Statistical relationships further suggest $B_{ext}$ can be used as a proxy of interannual fire emission variability in lowland

Bolivia. Significant statistical relationships were found between mean-monthly $B_{ext}$ and monthly MODIS C6 fire data from 2001–2015 (Figure 2), between mean-monthly $B_{ext}$ and mean-monthly GFWED variables from 2001–2015 (Table 2), and between mean-fire season $B_{ext}$ and mean-fire season GFWED variables from 1982–2015 (Table 3). Of the GFWED variables, seasonal covariation was strongest between $B_{ext}$ and DC from 2001–2015 (Fig. 3a, e). Interannual covariation was strongest between $B_{ext}$ and DC from 1982–2015 (Fig. 6d). Both seasonal and interannual covariation between $B_{ext}$ and DC

suggest visibility related to smoke emissions in lowland Bolivia was influenced by seasonal-to-interannual DC variability impact on fire activity. From 1982–2015, the strongest correlations (Table 3) were between mean-fire season CPC DC (i.e., August–October) and mean-fire season $B_{ext}$ (i.e., August–October). The next strongest correlation was between mean-fire season MERRA2 relative humidity and mean-fire season $B_{ext}$. Our results suggest from 2001–2015, and from 1982–2015, regional fire activity in Bolivia was affected by interannual DC and relative humidity variability.

### 4.4 Considerations and implications of GFWED and WMO-visibility data

We speculate antecedent dry conditions linked to precipitation and temperature anomalies prior to the fire season impacted high DC values and fire in Bolivia and NK. When southern Amazon wet season drought is severe terrestrial water storage deficits can amplify drought and fire severity during the subsequent dry season (Chen et al., 2013a). Wet season

drought leading to drought and fire seems plausible considering the DC is used here as an indicator of heavy-surface fuel drying over several months and deep, organic soil moisture content (Field et al., 2015). Further, this would explain weaker

linear correlations observed among fire, $B_{ext}$, precipitation and temperature. We recommend future studies investigating climate and fire relationships in tropical and subtropical ecosystems use the DC as an indicator of antecedent dry conditions.

Knowing the importance of DC as an indicator of antecedent dry conditions that influence fire and visibility (e.g., $B_{ext}$),
future studies using DC should consider various precipitation sources when calculating DC. DC values were less biased when using TRMM and GPCP precipitation, and demonstrated a strong correlation with MODIS C6 fire data for lowland Bolivia and NK (Table 1). MERRA2 DC and CPC DC values were consistently higher than TRMM DC and GPCP DC values for both lowland Bolivia and NK from 2001–2015. MERRA2 DC values exceeded 800 for lowland Bolivia and 1,000 for NK, much higher than TRMM and GPCP values.

In addition to the GFWED data used in our study, errors associated with the WMO-visibility data are important to consider. Visibility data are vulnerable to errors related to human-observed measurements, and are derived from spatially inconsistent weather stations distributed in the Amazon region (van Marle et al., 2017). Because of the role variable smoke transport has on relationships between fire activity and visibility, we have limited our visibility to only the broadest relationships across the large lowland area of Bolivia. We note, however, that the regional visibility signal was relatively
insensitive to whether it was calculated from 4, 6, 8, or 11 stations (Fig. A1).

Given these limitation, our results demonstrate (i.e., 2001–2015) connections among fire in lowland Bolivia, $B_{ext}$ variability (Fig. 2), interannual climate variability (Fig. 3, 4, 6), biome type (e.g., Fig. 4), forest loss (Fig. 5), and biome-boundary dynamics (e.g., Fig. 5). Future climate changes could impact drought severity and fire activity in lowland Bolivia. From 1979–1996, fire season length decreased, or did not change in lowland Bolivia (Jolly et al., 2015). However, from
1996–2013, fire season length increased in the Brazilian Amazon north and east of lowland Bolivia (Jolly et al., 2015), corresponding to a period of increased emissions in the southern Amazon (van Marle et al., 2017) and lowland Bolivia (Fig. 6). In the Amazon Basin, a projected increase in Fire Weather Index (FWI) is expected for period 2026–2045 (Bedia et al., 2015), and fire season severity is expected to increase during the 21[st] century (Flannigan et al., 2013). Projections of fire season length increasing, FWI increasing, and fire severity increasing (Flannigan et al., 2013) are of concern for lowland
Bolivia when considering our results, and the impacts of drought conditions have on increased fire activity in lowland Bolivia. A drier climate and associated fire in the Amazon could promote a transition from seasonally-inundated wetlands to

savannah vegetation, which could allow savanna forest expansion into the tropical Amazon and create an environment more susceptible to fire (Flores et al., 2017). Our results (Fig. 4) and the results of others (e.g., Flores et al., 2017) indicate seasonally-inundated wetlands and cerrado forests are vulnerable to fire associated with drought, suggesting these biomes

need to be carefully considered if drought in the Amazon occurs more frequently in the future. We provide further understanding of how different biomes have recently responded to drought and fire in lowland Bolivia, important when considering uncertainties regarding the fate of the Amazon (Zhang et al., 2015).

While increased FWI and fire severity are a concern for lowland Bolivia and for carbon emissions and global climate, fire leading to forest loss in the METF biome within NK was not observed from 2001–2015 (Fig. 2b, 4, 5). Our results

suggest if human activities that amplify fire in the southern Amazon were restricted, recent fire activity could have been reduced in the METF biome. Considering the spatial distribution of fires in NK, and the spatial coherence of forest loss and fire in the unprotected METF biome outside of NK (Fig. 5), a major limitation of our study is that we did not quantify the amount of forest loss in lowland Bolivia from human activities. As mentioned by others (e.g., Bedia et al., 2015), to better understand potential impacts of fire to southern Amazon tropical forests, human activities causing forest loss and fire need to

be considered. To minimize deforestation and fire in the southern Amazon, our results (e.g., Fig 5) and others (Flannigan et al., 2013) suggest human ignitions need to be reduced. Considering deforestation in the Brazilian Amazon has increased since 2012 (Fearnside, 2017), and in the southern Amazon cerrado biome since 2010 (Soares-Filho et al., 2014), land use incentives and agricultural policies implemented to reduce deforestation in parts of the Brazilian Amazon (Nepstad et al., 2014) need to be enforced throughout the Amazon. In lowland Bolivia, if land use incentives and agricultural policies to

reduce deforestation are not implemented and enforced, and the demand for Amazon resources continues to increase (Fernside, 2017), future anthropogenic deforestation and fire could worsen, particularly when drought occurs (e.g., Fig 4a, c).

## 5   Conclusions

We have demonstrated how multiple data can be used to explore seasonal and interannual relationships between climate, fire, land use, forest loss, and smoke emissions. A key finding, high DC and low humidity were dominant causes of recent

fire activity in unprotected and protected areas of lowland Bolivia. In addition, fire was likely enhanced by fragmented

biomes because of human activities, seen as forest loss in our results. Of interest to biogeographers, fires in NK from 2001–

2015, occurred primarily in the cerrado biome and in seasonally-inundated wetlands, and were absent from the NK METF

biome with the exception of cerrado–METF biome interfaces. Considering fire was minimal in the NK METF biome from

2001–2015, we recommend tropical forests in the southern Amazon and lowland Bolivia need further protection from human

ignitions and deforestation. Further, considering cerrado and seasonally-inundated wetlands susceptibility to fire when

drought occurs, attention should be given to cerrado expansion into seasonally-inundated wetlands and METF biomes.

In addition to exploring climate, fire, land use and biome relationships, our results demonstrate how differences between

precipitation estimates used to calculate DC, can bias DC values (e.g., MERRA2, CPC, GPCP and TRMM). Differing DC

values because of precipitation estimate uncertainties demonstrate the importance of using multiple data sources when

considering relationships among climate, fire, land use, forest loss, and smoke emissions. By using multiple data sources, we

were able to extend the historical fire record for lowland Bolivia using $B_{ext}$ visibility data. Our results and the results of

others suggest visibility data can be used as a proxy of regional fire emissions in the southwestern Amazon and lowland

Bolivia. Based on our results and the results of others, we recommend future studies interested in extending regional fire

records should consider using multiple data sources including MODIS active fire, GFWED, WMO-visibility data.

**Data Availability**

MODIS C6 data can be obtained at https://earthdata.nasa.gov/earth-observation-data/near-real-time/firms/active-fire-data.

GFWED data is available at https://data.giss.nasa.gov/impacts/gfwed/. Data used to calculate horizontal visibility can be

obtained from https://catalog.data.gov/dataset/integrated-surface-global-hourly-data. MODIS-based Global Land Cover

Climatology data is available at https://landcover.usgs.gov/global_climatology.php. Global Forest Change Landsat data can

be found at https://earthenginepartners.appspot.com/science-2013-global-forest/download_v1.2.html.

**Acknowledgments**

This material is based upon work supported by the National Science Foundation Graduate Research Fellowship under Grant

1256065. Any opinion, findings, and conclusions or recommendations expressed in this material are those of the authors(s)

and do not necessarily reflect the views of the National Science Foundation. RF was supported by the NASA Precipitation Measurement Missions Science Team and the NASA Modeling, Analysis and Prediction Program.

**Appendices**

Figure. A1

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

**Table 1.** Mean monthly Pearson's correlations (01/2001 – 12/2015) between Moderate Resolution Imaging Spectroradiometer 6 (MODIS C6) active fires ≥ 90% confidence, and Global Fire WEather Database (GFWED) variables for Bolivia (i.e. table columns 1-2) and NK (i.e. table columns 3-4). All correlation p-values were < .001 with 178 degrees of freedom, unless otherwise noted (e.g., NK MODIS and MERRA2 precipitation: p-value = .003). Correlations are listed in order of strongest (i.e. top row) to weakest (i.e. bottom row).

| Bolivia | 95% Confidence Interval | NK | 95% Confidence Interval |
|---|---|---|---|
| Bolivia MODIS & GCPC DC | 0.69 – 0.82 | NK MODIS & TRMM DC | 0.39 – 0.61 |
| Bolivia MODIS & TRMM DC | 0.68 – 0.81 | NK MODIS & GCPC DC | 0.38 – 0.60 |
| Bolivia MODIS & CPC DC | 0.65 – 0.79 | NK MODIS & CPC DC | 0.33 – 0.56 |
| Bolivia MODIS & MERRA2 DC | 0.58 – 0.74 | NK MODIS & MERRA2 DC | 0.28 – 0.53 |
| Bolivia MODIS & MERRA2 relative humidity | -0.56 – -0.73 | NK MODIS & MERRA2 temperature | 0.24 – 0.49 |
| Bolivia MODIS & MERRA2 temperature | 0.43 – 0.64 | NK MODIS & MERRA2 relative humidity | -0.21 – -0.47 |
| Bolivia MODIS & MERRA2 precipitation | -0.27 – -0.51 | NK MODIS & MERRA2 precipitation (*p-value = 0.003) | -0.07 – -0.35 |
| Bolivia MODIS & GCPC precipitation | -0.25 – -0.50 | NK MODIS & GCPC precipitation (*p-value = 0.004) | -0.07 – -0.35 |
| Bolivia MODIS & TRMM precipitation | -0.24 – -0.50 | NK MODIS & TRMM precipitation (*p-value = 0.004) | -0.06 – -0.34 |
| Bolivia MODIS & CPC precipitation | -0.24 – -0.49 | NK MODIS & CPC precipitation (*p-value = 0.01) | -0.05 – -0.33 |

**Table 2.** Mean monthly Pearson's correlations (01/2001 – 12/2015) between Bext(km-1), and Global Fire WEather Database (GFWED) variables for Bolivia (i.e., table columns 1-2), compared to the same Pearson's correlations over the entire GFWED record from 01/1982 to 12/2015 (i.e., table columns 3-4). All correlation p-values were < 0.001, unless otherwise noted (e.g., Bext(km-1) and MERRA2 precipitation: *p-value = 0.0013). Bolivia (01/2001 – 12/2015) correlations have 178 degrees of freedom, and Bolivia (01/1982 – 12/2015) correlations have 406 degrees of freedom


respectively. Correlations (i.e., table columns 1-2) are listed in order of strongest (i.e. top row) to weakest (i.e. bottom row).

| Bolivia (01/2001 – 12/2015) | 95% Confidence Interval | Bolivia (01/1982 - 12/2015) | 95% Confidence Interval |
|---|---|---|---|
| Bext(km$^{-1}$) & TRMM DC | 0.38 – 0.60 | N/A | N/A |
| Bext(km$^{-1}$) & GCPC DC | 0.37 – 0.60 | N/A | N/A |
| Bext(km$^{-1}$) & CPC DC | 0.34 – 0.57 | Bext(km$^{-1}$) & CPC DC | 0.45 – 0.59 |
| Bext(km$^{-1}$) & MERRA2 DC | 0.33 – 0.56 | Bext(km$^{-1}$) & MERRA2 DC | 0.36 – 0.52 |
| Bext(km$^{-1}$) & MERRA2 relative humidity | -0.24 – -0.49 | Bext(km$^{-1}$) & MERRA2 relative humidity | -0.38 – -0.53 |
| Bext(km$^{-1}$) & MERRA2 temperature | 0.24 – 0.49 | Bext(km$^{-1}$) & MERRA2 temperature | 0.33 – 0.49 |
| Bext(km$^{-1}$) & MERRA2 precipitation (*p-value = 0.0013) | -0.09 – -0.37 | Bext(km$^{-1}$) & MERRA2 precipitation | -0.16 – -0.35 |
| Bext(km$^{-1}$) & GCPC precipitation (*p-value = 0.004) | -0.07 – -0.35 | N/A | N/A |
| Bext(km$^{-1}$) & TRMM precipitation (*p-value = 0.005) | -0.07 – -0.35 | N/A | N/A |
| Bext(km$^{-1}$) & CPC precipitation (*p-value = 0.007) | -0.06 – -0.34 | Bext(km$^{-1}$) & CPC precipitation | -0.17 – -0.35 |



**Table 3.** Mean-fire season (i.e., August–October) Pearson's correlations (01/1982–12/2015) between Bext(km-1), and Global Fire WEather Database (GFWED) variables for lowland Bolivia. All correlation p-values were < 0.001, unless otherwise noted.

| Bolivia | 95% Confidence Interval |
|---|---|
| $B_{ext}(km^{-1})$ & CPC DC | 0.36 – 0.79 |
| $B_{ext}(km^{-1})$ & MERRA2 relative humidity | -0.25 – -0.74 |
| $B_{ext}(km^{-1})$ & MERRA2 DC (*p-value = 0.007) | 0.13 – 0.68 |
| $B_{ext}(km^{-1})$ & MERRA2 precipitation (*p-value = 0.014) | -0.10 – -0.66 |
| $B_{ext}(km^{-1})$ & CPC precipitation (*p-value = 0.046) | -0.008 – -0.61 |
| $B_{ext}(km^{-1})$ & MERRA2 temperature (*p-value = 0.13) | -0.08 – 0.55 |


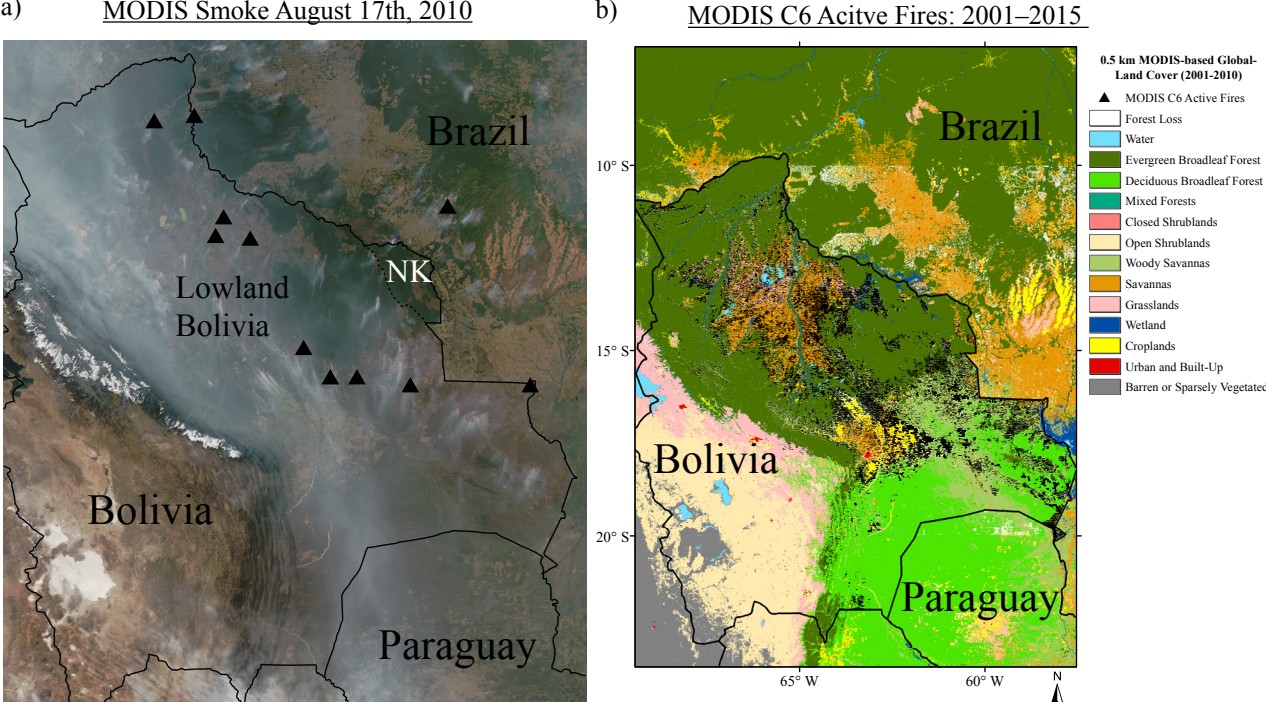

**Figure 1.** Fires and smoke observed by Moderate Resolution Imaging Spectroradiometer (MODIS) over Bolivia and Noel Kempff Mercado National Park (NK) on August 17th, 2010 (a), the highest fire year observed by the MODIS active fire product for Bolivia during the record (2001–2015). The locations of the eleven World Meteorological Organization (WMO)-level surface stations used to obtain visibility data are shown as black triangles (a). NASA image courtesy: Jeff Schmaltz. Forest loss from 2000–2012 (Hansen et al., 2013) displayed as white (b), and Moderate Resolution Imaging Spectroradiometer C6 (MODIS C6) active fires ≥ 90% confidence from 2001–2015, displayed as black triangles (b). MODIS based Collection 5.1 MCD12Q global land cover data [Broxton et al., 2014] is included (b).

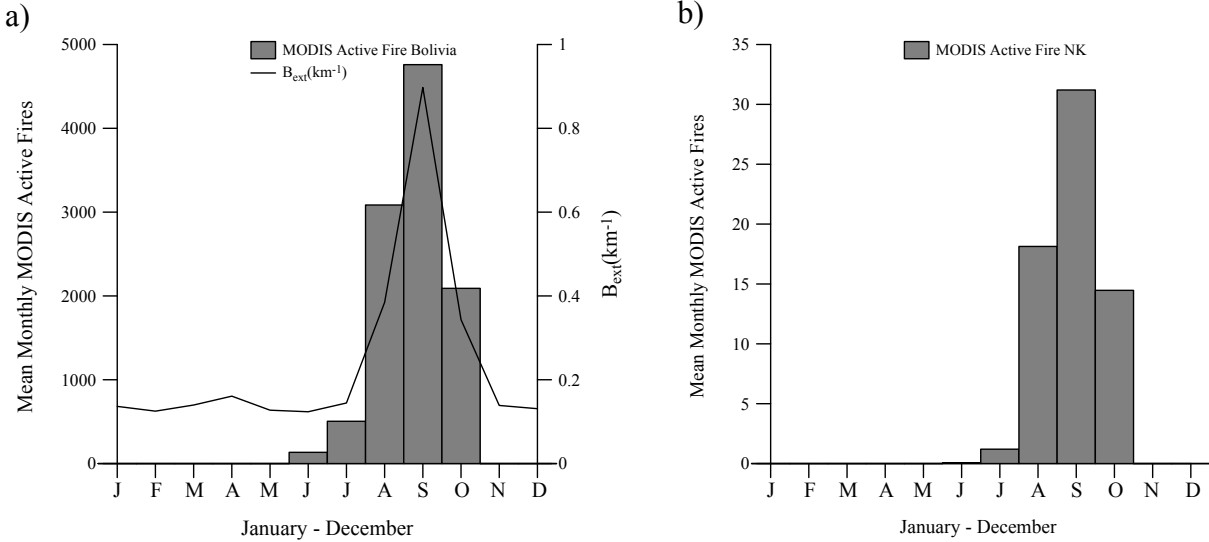

**Figure 2.** Mean-monthly Moderate Resolution Imaging Spectroradiometer 6 (MODIS 6) active fires ≥ 90% confidence (2001–2015) for Bolivia (a) and Noel Kempff Mercado National Park (NK) (b). Mean-monthly extinction coefficient ($B_{ext}$km$^{-1}$) (1973–2015) is included for Bolivia (a). Fire seasonality is clearly demonstrated for both Bolivia (a) and NK (b) (e.g. peak fire from August–October). For Bolivia, 85% of fires were detected from August–October (MODIS 6 active fire ≥ 90% confidence). For NK, 96% of fires were detected from August–October (MODIS 6 active fire ≥ 90% confidence). From 2001–2015, a 95% correlation confidence interval of 0.76 – 0.86, was observed between mean-monthly $B_{ext}$km$^{-1}$ and monthly lowland Bolivia MODIS C6 active fires (a).

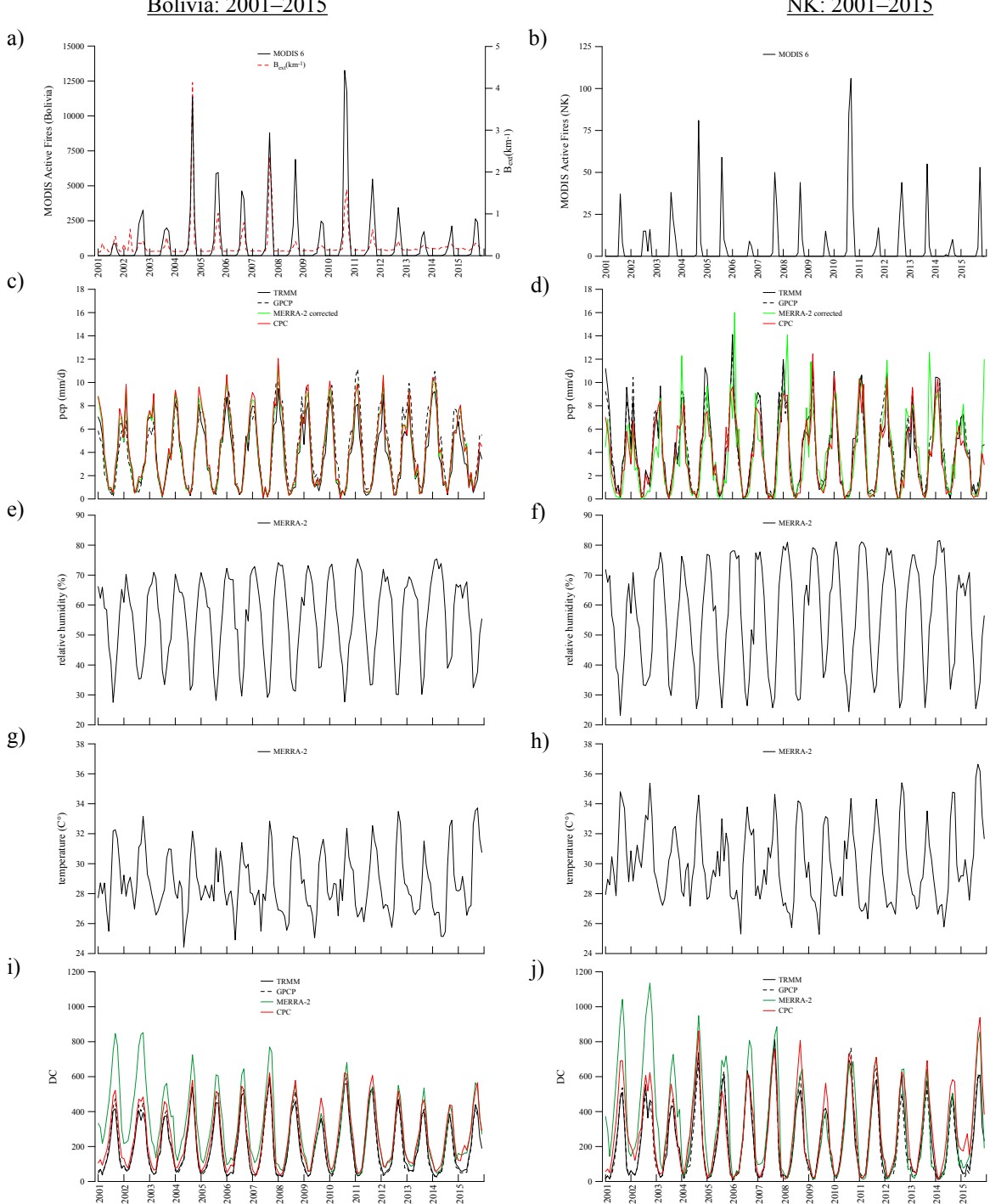

**Figure 3.** Mean-monthly (Jan–Dec) timeseries of MODIS C6 active fires (a,b), of the extinction coefficient Bext (a), and of selected Global Fire WEather Database (GFWED) variables (c–j). Timeseries are for Bolivia (a,c,e,g,i) and NK (b,d,f,h,j) from 2001–2015. GFWED variables include precipitation from four sources (c,d), MERRA-2 relative humidity (e,f), MERRA-2 temperature (g,h), and the drought code (DC) calculated from four sources (i,j).

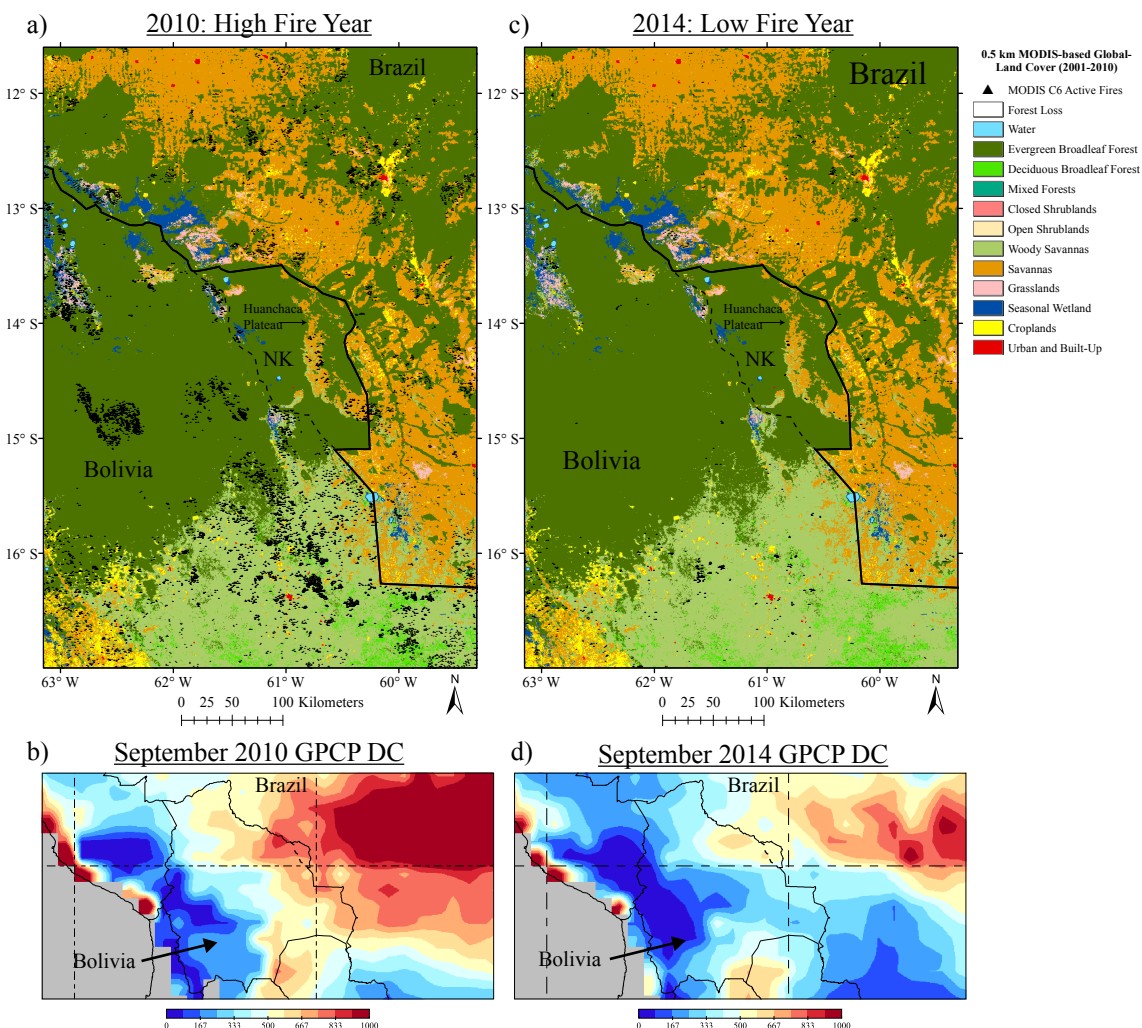

**Figure 4.** Moderate Resolution Imaging Spectroradiometer C6 (MODIS C6) active fires ≥ 90% confidence during 2010, a high fire year (a), and during 2014, a low fire year (c). September GPCP drought code (DC) during 2010 (b) and 2014 (d). MODIS based Collection 5.1 MCD12Q global land cover data [Broxton et al., 2014] is included. Noel Kempff Mercado National Park (NK) is the area that falls within the dotted-black polyline and the Bolivia-Brazil border.

MODIS C6 Acitve Fires: 2001 - 2015 and Forest Loss:
2000 - 2012

**Figure 5.** Forest loss from 2000 - 2012 (Hansen et al., 2013) displayed as white (a), and Moderate Resolution Imaging Spectroradiometer C6 (MODIS C6) active fires ≥ 90% confidence from 2001 - 2015 (b). MODIS based Collection 5.1 MCD12Q global land cover data [Broxton et al., 2014] is included. NK is the area that falls within the dotted-black polyline and the Bolivia-Brazil border.

## Lowland Bolivia 1982–2015

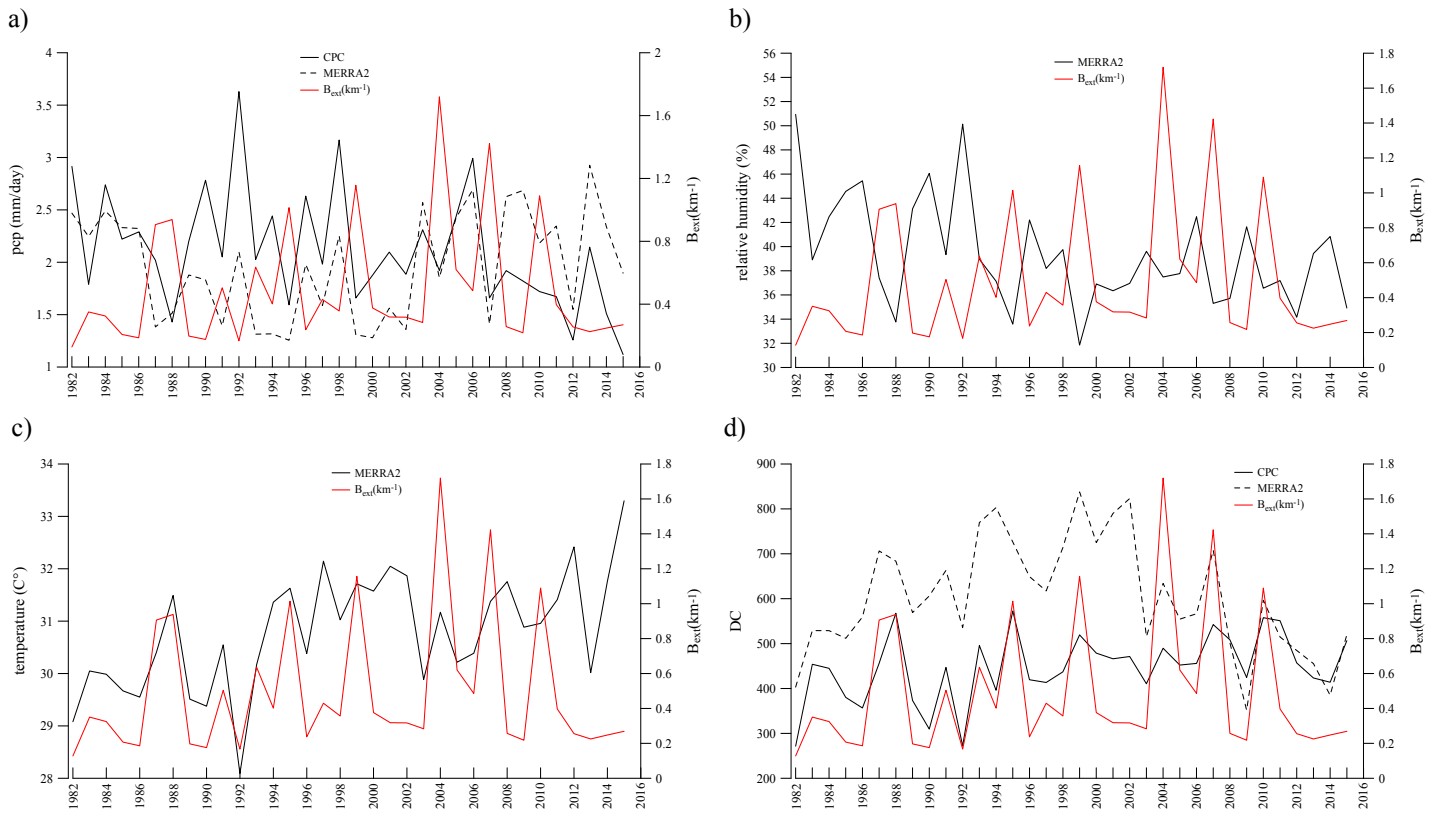

**Figure 6.** Mean fire season (Aug–Oct) time series (1982–2015) of daily Global Fire WEather Database (GFWED) variables for Bolivia including MERRA2 precipitation (a), MERRA2 relative humidity (b), MERRA2 temperature (c), and CPC and MERRA2 DC (d). $B_{ext}$(km$^{-1}$) is included in each plot (i.e., red line).