# Peer review of "The impacts of recent drought in lowland Bolivia on fire, forest loss and regional smoke emissions"

_Biogeosciences, 2017_

## Referee Comment (RC1) · Anonymous Referee #1 · 22 Feb 2018

General comments: The authors of this manuscript presented a comprehensive analysis on the relationships between drought, fire, and smoke, using multiple sources of observations. The topic is interesting and relevant to the scope of Biogeosciences. The results have values for both atmosphere and land communities. However, the authors need to address several major issues described below.

Specific comments:

1. Correlation analyses for different time scales need to be separated and clarified. The authors performed pearson's correlation tests for two types of data: mean-monthly data and mean-fire season data. The correlations from mean-monthly data are mostly

from the covariation in seasonal cycles, while the correlations from mean-fire season data are mainly from interannual variability. However, these two types of analyses are not clearly differentiated, sometimes they are even directly compared in the manuscript. This leads to unnecessary confusions and incorrect interpretations.

For example, in Page 9, line 212-214, the authors state "Nevertheless, many relationships were significant. Based on these relationships, Bext visibility data is used as a proxy of longer-term (i.e., 1982–2015) regional fire activity in lowland Bolivia'. The first sentence is referring to the correlations between monthly Bext and MODIS variables. I don't think these correlations validate the assumption that 'Bext can be used as a proxy of long-term regional fire activity'.

Page 9, line 233-234: What are exactly the 'normal's mean in 'lower-than-normal' and 'higher-than-normal'? A logical guess is that the 'normal' refers to the monthly climatology, and this sentence is about explaining the interannual variability. But this will be contrary to Table 1 cited the later part of the sentence, which is all about seasonal variability.

There are several other occasions of such confusions not listed above. I suggest the authors go over the manuscript and clear all cases.

2. There's something wrong in Figure 5. In several places of the text, you talked about Fig. 5a and Fig. 5b (e.g., in Page 8, line 200-201). But I only see one panel in Figure 5. Since I couldn't locate the positions for forest loss (white color pixels?) in Figure 5, I am basically unable to review the whole section of 3.5, as well as the first paragraph in section 4.1.

Technical corrections:

Page 2, line 48-50: This sentence is difficult to understand. Page 3, line 66-69: Again, I don't quite understand this sentence. Please rephrase it. Page 3, line 76: The spatial resolution of MODIS active fire product should be 1km, not 500 meters. Page 5, line

121: Better to change '13°S x 15.3°S, 62.2°W x 59.5°W' to '13°S - 15.3°S, 62.2°W - 59.5°W'. Page 7, line 175: 'Fig. 3f' is about MERRA2 data, not MODIS data. Page 8, line 208: Where did you show this: "the positive relationship between lowland Bolivia MODIS C6 active fire data and mean-September Bext"? Page 20, Figure 1: You combined these biomes into several groups "cerrado, SDTF, METF and seasonally inundated wetland biomes" in the text, and discussed your results mainly based on the group classifications. Why didn't you the grouped vegetation types in Figures 1 and 4?

---

## Author Comment (AC1) · 2 Mar 2018

The comments and concerns made by referee 1 are greatly appreciated and will be taken into consideration in the updated manuscript. Below is a summary list of referee 1 main comments and concerns, and how they will be addressed in the updated manuscript draft.

REFEREE SPECIFIC COMMENTS 1-4:

REFEREE COMMENT 1. Lack of clarity between mean-monthly correlations and mean-fire season correlations.

[Figure]

REFEREE COMMENT 2. Pearson's correlations do not validate that Bext visibility data can be used as a proxy of fire activity for lowland Bolivia

AUTHORS RESPONSE TO COMMENTS 1 & 2:

- In consideration of both concerns, we have made the following changes:

- We describe in greater detail the correlations that were performed in the Methods section "2.5 Statistical Analysis". The Method section now clearly states which correlations are mean-monthly, and which are mean-fire season. In addition, in the Results and Discussion sections we make clear which correlations are mean-monthly and which are mean-fire season. Further, the referee's comment that mean-monthly correlations are representative of seasonal variability and the mean-fire season correlations are representative of interannual variability were taken into consideration and included in our Results and Discussion sections.

- We changed 3.6 and 4.3 subsection titles. The titles were changed to more clearly present statistical relationships in section 3.6, and save any discussion of the statistical relationships for section 4.3. We maintain that our results are consistent with Marle et al. (2017), as well as other studies that have used horizontal visibility data to extend regional fire records (Field et al., 2009; Field et al., 2016). Thus, with caution we suggest Bext is related to fire activity prior to the MODIS record. Further, a strong correlation is established between lowland Bolivia MODIS active fire and Bext seasonality in Figure 2 in our paper (2001 - 2015), indicating lowland Bolivia fire and Bext seasonal variability are highly correlated. Given our data is consistent with others, reproducible, and the strong statistical relationships are established, we suggest past Bext variability is related to past fire activity in lowland Bolivia prior to the MODIS record.

REFEREE COMMENT 3.

Page 9, line 233-234: What are exactly the 'normal's mean in 'lower-than-normal' and 'higher-than-normal'? A logical guess is that the 'normal' refers to the monthly clima-

tology, and this sentence is about explaining the interannual variability. But this will be contrary to Table 1 cited the later part of the sentence, which is all about seasonal variability.

AUTHORS RESPONSE TO COMMENT 4:

- Because our study is an investigation of how relationships between fire, climate, and visiblity (i.e., smoke) vary interannually and seasonally, we have removed any 'lower-than or higher-than-normal' interpretations. Instead, we discuss fire, climate and visibility data values as high or low relative to other values during that year, or between years (e.g., 2004 and 2010 were high fire years compared to the other 13 years).

REFEREE COMMENT 4.

There's something wrong in Figure 5. In several places of the text, you talked about Fig. 5a and Fig. 5b (e.g., in Page 8, line 200-201). But I only see one panel in Figure 5. Since I couldn't locate the positions for forest loss (white color pixels?) in Figure 5, I am basically unable to review the whole section of 3.5, as well as the first paragraph in section 4.1.

AUTHORS RESPONSE TO COMMENT 4:

- Figure 5a was not included in this updated manuscript version. I forgot to change all of the Fig. 5a and 5b to Fig. 5. The updated manuscript refers to Fig. 5 throughout. We thank the referee for making us aware of this mistake.

- As for the forest loss "white pixels", because Figure 5 is a high-resolution figure, if you zoom in you can see the white color pixels (i.e., positions of forest loss). A sentence was added to make this is made clear in the Methods section.

REFEREE TECHNICAL CORRECTIONS AND AUTHORS RESPONSE:

We thank the referee for taking their time in making technical corrections on our manuscript. These corrections have improved our manuscript, and are much appreciated. Thank you.

Page 2, line 48-50: This sentence is difficult to understand. Page 3, line 66-69: Again, I don't quite understand this sentence. Please rephrase it. - This sentence will be rephrased for clarity in the updated manuscript.

Page 3, line 76: The spatial resolution of MODIS active fire product should be 1km, not 500 meters. - Resolution changed to 1km

Page 5, line 121: Better to change '13âŮęS x 15.3âŮęS, 62.2âŮęW x 59.5âŮęW' to '13âŮęS - 15.3âŮęS, 62.2âŮęW - 59.5âŮęW'. - Changed to recommended text

Page 7, line 175: 'Fig. 3f' is about MERRA2 data, not MODIS data. - The Fig. 3 labels that were incorrectly referenced in the text were corrected in the updated manuscript.

Page 8, line 208: Where did you show this: "the positive relationship between lowland Bolivia MODIS C6 active fire data and mean-September Bext"? - Relationship should be for the fire season (August - October). The text was changed to "mean-fire season (i.e., August – October). The relationship is shown in Figure 2.

Page 20, Figure 1: You combined these biomes into several groups "cerrado, SDTF, METF and seasonally inundated wetland biomes" in the text, and discussed your re-sults mainly based on the group classifications. Why didn't you the grouped vegetation types in Figures 1 and 4? - This was done for simplicity when discussing in-text the fire-biome spatial analyses. The land classification could have been further simplified visually, but the details between the many different biomes would be lost to readers interested in higher level of detail. - I made this clearer in the methods section 2.1

---

## Referee Comment (RC2) · Anonymous Referee #3 · 27 May 2018

The authors use several data sources to study the controlling factors on interannual fire variability in lowland Bolivia considering in their analyses protected (Noel Kempff Mercado National Park) and unprotected areas as well as different biomes and vegetation types.

However, the vegetation types are poorly described just using MODIS land cover data without any ecological background. For example, what kind of deciduous needleleaf forest occur in this part of the Amazon forest which is dominated by angiosperms? It is not clear to which kind of wetlands the authors refer to. While in the text wetlands are characterized as seasonally-flooded types they are indicated as permanently flooded

in figures 1, 4, and 5.

I miss a discussion on the main triggers causing the severe droughts in the Amazon. Reduced rainfall, higher-than-normal temperatures, and reduced atmospheric moisture during the wet and dry seasons are mainly caused by sea surface temperature anomalies in the tropical Atlantic and the Equatorial Pacific. This should be addressed in the discussion citing relevant literature.

Recently, some papers discuss the vulnerability of different intact forest ecosystems (floodplains and non flooded forests) to wildfires in the Amazon (i.e., Flores et al. 2017 and studies cited in this paper), which also should be addressed in the discussion of the observed results.

In many studies the spatial patterns of annual maximum cumulative water deficit (MCWD) during severe droughts are used to explain the consequent enhancement of active fire incidence. MCWD is a useful indicator of meteorologically induced water stress without taking into account local soil conditions and plant adaptations, which are poorly understood in Amazonia. Why did the authors not use this proxy to relate fire occurrence in the study region?

The authors do not discuss the relationship between heavy smoke from forest fires in the Amazon and the regional precipitation regime. Andreae et al. (2004), for instance, observed that smokes from wildfires in the Amazon result into a reduced cloud droplet size causing a delay of the onset of precipitation. Pyro-clouds cause a suppression of low-level rainout and aerosol washout and allows the transport of water and smoke to upper levels causing intense thunderstorms. These clouds, attaining the stratosphere, have profound radiative impacts on the climate system (see also Koren et al. 2008).

Minor concerns:

L. 19-20: I think it is the other way round: Bext visibility data are linked to the interannual Drought Code (DC), as the emission of aerosols is a consequence of anthropogenic

fires favoured by droughts. Please indicate the meaning of Bext the first time you indicate it.

L. 37-39: Recently deforestation rates increased again in the Brazilian Amazon, particularly in the Southern Amazon region.

L. 40-42: The authors should also refer to the severe drought in 2010 (Lewis et al. 2010, Aragão et al. 2018) which had much broader impacts in the Southern Amazon than the drought of 2005 which was spatially restricted to the SW-Amazon.

L. 90/91: The Amazon is dominated by angiosperms. What kind of deciduous needle-leaf forest are those?

L. 92/93: This sentence has a contradiction as the authors mention seasonally inundated wetlands and refer to permanent wetland types. See the paper of Junk et al. (2011) on the classification of Amazonian wetlands and provide a better description of these ecosystems in the studied region.

L. 229: For my knowledge the year of 2005, not 2004, was a severe drought year affecting this particular region (Lewis et al. 2010, Aragão et al. 2018). The year of 2007 was an El Niño Year, in 2010 El Niño and especially increased SST anomalies in the Northern Atlantic caused the severe drought in the Southern Amazon basin.

L. 311: It is not clear whether fire season length from 1979–1996 decreased, or didn't change in lowland Bolivia. Are these observation based on different studies? However, only one study is cited (Jolly et al., 2015).

L. 328-330: Recent incentives and policies implemented in Brazil (revisions of its Forest Code) led to an increase of deforestation rates in recent years. Please specify the incentives and policies.

The scales of figures 4 c,d are difficult to read.

Corrections: L. 125, 152, 154, 157, 174: Units should be consistently written in the

exponential form (e.g., mm day–1)

L. 317: Insert "of" between "impacts" and "drought conditions".

---

## Author Comment (AC2) · 5 Jun 2018

Dear reviewer,

Thank you for your comments and suggestions (Referee #3 comment). They have been shared with the coauthors, and will be considered in the updated manuscript. Below you will find responses to your comments, with the changes that will be made in the updated manuscript. Once again, we thank you for the useful comments that will help improve our manuscript.

Referee #3 comment: The authors use several data sources to study the controlling

factors on interannual fire variability in lowland Bolivia considering in their analyses protected (Noel Kempff Mercado National Park) and unprotected areas as well as different biomes and vegetation types. However, the vegetation types are poorly described just using MODIS land cover data without any ecological background. For example, what kind of deciduous needleleaf forest occur in this part of the Amazon forest which is dominated by angiosperms? It is not clear to which kind of wetlands the authors refer to. While in the text wetlands are characterized as seasonally-flooded types they are indicated as permanently flooded in figures 1, 4, and 5.

RESPONSE: The reviewer is correct. The Amazon is comprised and dominated by angiosperms. The inclusion of needleleaf forest was an error in the text and in the legend in Figs 4 and 5. To determine if there are deciduous needleleaf forest in lowland Bolivia a query search for the landcover classification was performed. The land cover type was not found. Therefore, needleleaf forest has been removed from the legends and in-text.

According to Gosling et al. (2005) and Junk et al., (2011), wetlands in the riparian corridors of Noel Kempff Mercado National Park are seasonally flooded savannas. The legend is mislabeled in Figures 1, 4 and 5. The figure labels will be changed from "permanently flooded" to "seasonally flooded" to more accurately represent the land cover type. We will also review/include the Junk et al. (2011) paper on wetlands in the southern Amazon in the updated manuscript.

Referee #3 comment: I miss a discussion on the main triggers causing the severe droughts in the Amazon. Reduced rainfall, higher-than-normal temperatures, and reduced atmospheric moisture during the wet and dry seasons are mainly caused by sea surface temperature anomalies in the tropical Atlantic and the Equatorial Pacific. This should be addressed in the discussion citing relevant literature.

RESPONSE: Sea-surface temperature anomalies in the tropical Atlantic and Equatorial Pacific were mentioned in original manuscript drafts. However, to conserve word

count, and to focus our paper on smaller-spatial scale climate forcing, we omitted discussions on sea-surface temperature anomalies in this manuscript version. In the updated manuscript, we will include discussions on sea-surface temperature anomalies with relevant literature cited.

Referee #3 comment: Recently, some papers discuss the vulnerability of different intact forest ecosystems (floodplains and non flooded forests) to wildfires in the Amazon (i.e., Flores et al. 2017 and studies cited in this paper), which also should be addressed in the discussion of the observed results. In many studies the spatial patterns of annual maximum cumulative water deficit (MCWD) during severe droughts are used to explain the consequent enhancement of active fire incidence. MCWD is a useful indicator of meteorologically induced water stress without taking into account local soil conditions and plant adaptations, which are poorly understood in Amazonia. Why did the authors not use this proxy to relate fire occurrence in the study region?

RESPONSE: The MWD can be used for the reasons mentioned by the reviewer, and has been used to study fire and drought in the Amazon (e.g., Aragão et al., 2007). Using the MWD, a link has been identified between rainfall anomalies, drought and fire in the Amazon. While soil conditions and plant adaptations are poorly understood, and we could have used the MWD, our study used the drought code (DC) to better understand how net drying of deep fuels impact fire in the southwestern Amazon. Lower DC values were observed during the wet season and higher DC values during the dry season, consistent with other studies (Field et al., 2015). We use the DC as an indicator of antecedent dry (wet) conditions during the wet and dry seasons, which influence high (low) DC values during the following fire season from August–October in lowland Bolivia. Other indices and metrics could have been used. However, the DC used in our study captured the relationship between drought and fire in lowland Bolivia. Further, the DC was calculated from raw MERRA2 precipitation estimates and MERRA2 rain gauge corrected data to address uncertainties in our analyses.

Referee #3 comment: The authors do not discuss the relationship between heavy

smoke from forest fires in the Amazon and the regional precipitation regime. Andreae et al. (2004), for instance, observed that smokes from wildfires in the Amazon result into a reduced cloud droplet size causing a delay of the onset of precipitation. Pyro-clouds cause a suppression of low-level rainout and aerosol washout and allows the transport of water and smoke to upper levels causing intense thunderstorms. These clouds, attaining the stratosphere, have profound radiative impacts on the climate system (see also Koren et al. 2008).

RESPONSE: We will mention the impacts of smoke on regional precipitation in the Amazon in the updated manuscript were appropriate (e.g., section 4.3). Results from Andreae et al. (2004) are very interesting, and certainly are relevant to our work. Thank you for mentioning this research to us.

Minor concerns:

Referee #3 comment: L. 19-20: I think it is the other way round: Bext visibility data are linked to the interannual Drought Code (DC), as the emission of aerosols is a consequence of anthropogenic fires favoured by droughts.

RESPONSE: While we acknowledge the importance of anthropogenic fire and drought in our paper, we do not specifically measure anthropogenic ignitions. While other research has certainly shown your point above to be true, we simply are reporting our results. Our results identified a link between drought and fire, and between drought and visibility. We do several times mention the importance of anthropogenic fires in lowland Bolivia and cite relevant literature.

Referee #3 comment: Please indicate the meaning of Bext the first time you indicate it.

RESPONSE: An explanation of the Bext is included section 2.2. of Methods and Data.

Referee #3 comment: L. 37-39: Recently deforestation rates increased again in the Brazilian Amazon, particularly in the Southern Amazon region.

RESPONSE: We will include a sentence or two regarding the current state of deforestation in the southern Amazon in the updated manuscript. From reading the short letter from Fearnside (2017), it seems that economic forcing plays a significant role in deforestation and fire in the region. The first introductory paragraph was modified to reflect this. Fearnside, P. (2017). Business as usual: a resurgence of deforestation in the Brazilian Amazon. Yale Environ, 360.

Referee #3 comment: L. 40-42: The authors should also refer to the severe drought in 2010 (Lewis et al. 2010, Aragão et al. 2018) which had much broader impacts in the Southern Amazon than the drought of 2005 which was spatially restricted to the SW-Amazon.

RESPONSE: We will discuss the importance of the 2010 drought and cite relevant literature (e.g., Lewis et al. 2010) in the updated manuscript.

Referee #3 comment: L. 90/91: The Amazon is dominated by angiosperms. What kind of deciduous needle-leaf forest are those?

RESPONSE: The deciduous needle-leaf forests are not present in our area of analyses, and will be removed from the legends for Figures 4 and 5.

Referee #3 comment: L. 92/93: This sentence has a contradiction as the authors mention seasonally inundated wetlands and refer to permanent wetland types. See the paper of Junk et al.(2011) on the classification of Amazonian wetlands and provide a better description of these ecosystems in the studied region.

RESPONSE: The Junk et al. (2011) description of hydromorphic climate savannas found in lowland Bolivia will be included in the updated manuscript.

Referee #3 comment: L. 229: For my knowledge the year of 2005, not 2004, was a severe drought year affecting this particular region (Lewis et al. 2010, Aragão et al. 2018). The year of 2007 was an El Niño Year, in 2010 El Niño and especially increased SST anomalies in the Northern Atlantic caused the severe drought in the Southern Amazon basin.

[Figure]

RESPONSE: The year 2005 was the severe drought year mentioned by Chen et al. (2013b). L. 229 was changed from 2004 to 2005. However, our results (fig 3 I,j) suggest that 2004 was a significant drought year as well for lowland Bolivia.

Referee #3 comment: L. 311: It is not clear whether fire season length from 1979–1996 decreased, or didn't change in lowland Bolivia. Are these observation based on different studies? However, only one study is cited (Jolly et al., 2015).

RESPONSE: This information is from Figure 3b from Jolly et al. (2015). The sentences below have been changed for clarity. Fire season length decreased or did not change in lowland Bolivia from 1996–2013, compared to fire season length from 1979 – 1996 (Jolly et al., 2015). However, from 1996–2013, fire season length increased in the Brazilian Amazon north and east of lowland Bolivia, compared to fire season length from 1979 – 1996 (Jolly et al., 2015).

Referee #3 comment: L. 328-330: Recent incentives and policies implemented in Brazil (revisions of its Forest Code) led to an increase of deforestation rates in recent years. Please specify the incentives and policies.

RESPONSE: This section was modified to discuss recent deforestation in the southern Amazon. Soares-Filho et al., 2014 Nepstad et al., 2014 Fearnside, P. (2017). Business as usual: a resurgence of deforestation in the Brazilian Amazon. Yale Environ, 360.

Referee #3 comment: The scales of figures 4 c,d are difficult to read.

RESPONSE: The scales are small, but clear if "zoomed in" to read them. If the scales were larger, they would interfere with the other graphics in the figures.

Referee #3 comment: Corrections: L. 125, 152, 154, 157, 174: Units should be consistently written in the exponential form (e.g., mm day$^{-1}$)

RESPONSE: Corrections will be made in the updated manuscript.

Referee #3 comment: L. 317: Insert "of" between "impacts" and "drought conditions".

RESPONSE: Corrections will be made in the updated manuscript.

---

## Author Response (AR1)

REVIEWER 3:

Recently, some papers discuss the vulnerability of different intact forest ecosystems (floodplains and non flooded forests) to wildfires in the Amazon (i.e., Flores et al. 2017 and studies cited in this paper), which also should be addressed in the discussion of the observed results. In many studies the spatial patterns of annual maximum cumulative water deficit (MCWD) during severe droughts are used to explain the consequent enhancement of active fire incidence. MCWD is a useful indicator of meteorologically induced water stress without taking into account local soil conditions and plant adaptations, which are poorly understood in Amazonia. Why did the authors not use this proxy to relate fire occurrence in the study region?

ASSOCIATE EDITOR DECISION:

I missed in your author's response to reviewer 3 a statement how you will address in the discussion the vulnerability of different intact forest ecosystems in the Amazon (floodplains and non-flooded forests) to wildfires (i.e., Flores et al. 2017 and studies cited in this paper).

OUR RESPONSE:

We added the following sentences in our discussion to reflect the results of Flores (et al., 2017) and our work to address "the vulnerability of different intact forest ecosystems (floodplains and non flooded forests) to wildfires in the Amazon."

"A drier climate and associated fire in the Amazon could promote a transition from seasonally-inundated wetlands to savannah vegetation, which could allow savanna forest expansion into the tropical Amazon and create an environment more susceptible to fire (Flores et al., 2017). Our results (Fig. 4) and the results of others (e.g., Flores et al., 2017) indicate seasonally-inundated wetlands and cerrado forests are vulnerable to fire associated with drought, suggesting these biomes need to be carefully considered if drought in the Amazon occurs more frequently in the future."

The MWD can be used for the reasons mentioned by the reviewer, and has been used to study fire and drought in the Amazon (e.g., Aragão et al., 2007). Using the MWD, a link has been identified between rainfall anomalies, drought and fire in the Amazon. While soil conditions and plant adaptations are poorly understood, and we could have used the MWD, our study used the drought code (DC) to better understand how net drying of deep fuels impact fire in the southwestern Amazon. Lower DC values were observed during the wet season and higher DC values during the dry season, consistent with other studies (Field et al., 2015). We use the DC as an indicator of antecedent dry (wet) conditions during the wet and dry seasons, which influence high (low) DC values during the following fire season from August–October in lowland Bolivia. Other indices and metrics could have been used. However, the DC used in our study captured the relationship between drought and fire in lowland Bolivia. Further, the DC was calculated from

raw MERRA2 precipitation estimates and MERRA2 rain gauge corrected data to address uncertainties in our analyses.

A key finding from our study, "high DC and low humidity were dominant causes of recent fire activity in unprotected and protected areas of lowland Bolivia." We not only demonstrate how an indicator of meteorological induced water stress (i.e., low relative humidity) can enhance fire occurrence, we also demonstrate how drought affecting surface fuels enhances fire in the southern Amazon. Further, we found "In addition to high DC values regionally, low relative humidity (Fig. 3c) and high temperature (Fig. 3d) were linked to increased fire activity in lowland Bolivia and NK (Table 1). Further demonstrating the connection between drought and fire in NK and lowland Bolivia were drought years and non-drought years, when fire was either significantly enhanced (Fig. 4a), or reduced (Fig. 4c)." We clearly demonstrate links between metrological induced water stress and fire occurrence in our study. Further, we mention in the discussion section previous studies that have identified metrological induced water stress in the Amazon. For example, "
[revised manuscript text omitted]

variable smoke transport has on relationships between fire activity and visibility, we have limited our visibility to only the broadest relationships across the large lowland area of Bolivia. We note, however, that the regional visibility signal was relatively insensitive to whether it was calculated from 4, 6, 8, or 11 stations (Fig. A1).

Given these limitation, our results demonstrate (i.e., 2001–2015) connections among fire in lowland Bolivia, $B_{ext}$ variability (Fig. 2), interannual climate variability (Fig. 3, 4, 6), biome type (e.g., Fig. 4), forest loss (Fig. 5), and biome-boundary dynamics (e.g., Fig. 5). Future climate changes could impact drought severity and fire activity in lowland Bolivia. From 1979–1996, fire season length decreased, or did not change in lowland Bolivia (Jolly et al., 2015). However, from 1996–2013, fire season length increased in the Brazilian Amazon north and east of lowland Bolivia (Jolly et al., 2015), corresponding to a period of increased emissions in the southern Amazon (van Marle et al., 2017) and lowland Bolivia (Fig. 6). In the Amazon Basin, a projected increase in Fire Weather Index (FWI) is expected for period 2026–2045 (Bedia et al., 2015), and fire season severity is expected to increase during the 21$^{st}$ century (Flannigan et al., 2013). Projections of fire season length increasing, FWI increasing, and fire severity increasing (Flannigan et al., 2013) are of concern for lowland Bolivia when considering our results, and the impacts of drought conditions have on increased fire activity in lowland Bolivia. A drier climate and associated fire in the Amazon could promote a transition from seasonally-inundated wetlands to savannah vegetation, which could allow savanna forest expansion into the tropical Amazon and create an environment more susceptible to fire (Flores et al., 2017). Our results (Fig. 4) and the results of others (e.g., Flores et al., 2017) indicate seasonally-inundated wetlands and cerrado forests are vulnerable to fire associated with drought, suggesting these biomes need to be carefully considered if drought in the Amazon occurs more frequently in the future. We provide further understanding of how different biomes have recently responded to drought and fire in lowland Bolivia, important when considering uncertainties regarding the fate of the Amazon (Zhang et al., 2015).

While increased FWI and fire severity are a concern for lowland Bolivia and for carbon emissions and global climate, fire leading to forest loss in the METF biome within NK was not observed from 2001–2015 (Fig. 2b, 4, 5). Our results suggest if human activities that amplify fire in the southern Amazon were restricted, recent fire activity could have been reduced in the METF biome. Considering the spatial distribution of fires in NK, and the spatial coherence of forest loss and fire in the unprotected METF biome outside of NK (Fig. 5), a major limitation of our study is that we did not quantify the amount of forest loss in lowland Bolivia from human activities. As mentioned by others (e.g., Bedia et al., 2015), to better understand potential impacts of fire to southern Amazon tropical forests,

**Commented [JH7]:** Reworded for clarification

**Commented [JH8]:** The vulnerability of seasonally-inundated wetlands and savannah/cerrado to wildfire in the Amazon was added, as well as a reference to the Flores et al. (2017) paper mentioned.

[revised manuscript text omitted]

---

## Editor Decision (ED1)

**Technical corrections bg-2017-462**

L. 126: I suggest using "meteorological" instead of "weather".

L. 134: Use en dashes for the ranges of the coordinates.

L. 137/138: Please indicate clearly the temporal resolution of temperature and RH. Are these daily values or monthly averages? Is it mean or maximum temperature?

L. 157-160: Is suggest writing "Next, to provide information on seasonal relationships correlations were performed between monthly MODIS C6 data (i.e., total monthly fires) and mean-monthly GFWED data, as well as between mean-monthly WMO-visibility data and mean-monthly GFWED data, both for the period 01/2001–12/2015."

L. 172, 174, 177 and 195: fires year$^{-1}$

L. 239: Delete spaces after "1982" and before "1993" (1982–1993)

L. 248, 250, 251, 253, 290: Correct author's name is Aragão.

L. 251: there is no need to "b" for the reference (crosscheck with line 597); do not indicate "et al." in italic.

L. 254: Delete spaces (October/November–April/May)

L. 294, 352, 369, 381: Figs. (check also other parts of the manuscript with multiple citations of figures).

L. 339/340: I suggest to substitute "DC" by "parameter" at the end of this sentence avoiding the repetition of DC at the end and beginning of the following sentence.

L. 375: add a dot after Fig

References:

Again, listed and cited references in the text (papers of two authors) are not formatted according the norms of Biogeosciences (https://www.biogeosciences.net/Copernicus_Publications_Reference_Types.pdf)

Legend Table 1: 01/2001–12/2015

Legend Table 2: 01/1982–12/2015 and 01/2001–12/2015 (two times); correct "Weather"; put "-1" as uppercase.

Table 2: The lines of the table should be formatted to read the uppercase numbers (Bext km$^{-1}$); correct: 01/1982–12/2015 and 01/2001–12/2015; avoid underlining in table

Legend Table 3: put "-1" as uppercase number.

Table 3: The lines of the table should be formatted allowing to read the uppercase numbers (Bext km$^{-1}$)

Figures 3 and 6: Increase size of axis titles and numbers and legends (they are difficult to read in the printed version). Indicate precipitation (mm day$^{-1}$) at the title of the y-axis of panels 3c,d and 6a,b.

Figures 4a,b, and 5: Black color of words in the map are difficult to read, please use white color to increase the contrast.

Legend of figure 5: correct 2001–2015, why using [] for citation? (see also legend of figure 4).

Table S1: The lines of the table should be formatted to read the uppercase numbers (Bex km$^{-1}$).